# Advances in Preparation and Properties of Regenerated Silk Fibroin

**DOI:** 10.3390/ijms241713153

**Published:** 2023-08-24

**Authors:** Linlin Huang, Jifeng Shi, Wei Zhou, Qing Zhang

**Affiliations:** College of Sericulture, Textile and Biomass Sciences, Southwest University, Chongqing 400715, China

**Keywords:** silk fibroin fiber, regenerated silk fibroin, dissolution method, structure, property

## Abstract

Over the years, silk fibroin (SF) has gained significant attention in various fields, such as biomedicine, tissue engineering, food processing, photochemistry, and biosensing, owing to its remarkable biocompatibility, machinability, and chemical modifiability. The process of obtaining regenerated silk fibroin (RSF) involves degumming, dissolving, dialysis, and centrifugation. RSF can be further fabricated into films, sponges, microspheres, gels, nanofibers, and other forms. It is now understood that the dissolution method selected greatly impacts the molecular weight distribution and structure of RSF, consequently influencing its subsequent processing and application. This study comprehensively explores and summarizes different dissolution methods of SF while examining their effects on the structure and performance of RSF. The findings presented herein aim to provide valuable insights and references for researchers and practitioners interested in utilizing RSF in diverse fields.

## 1. Introduction

Silk fibroin (SF) is an organic polymer material derived from natural silk, possessing excellent mechanical properties [1,2], good biocompatibility [3], controllable biodegradability [4,5], and ease of modification [6]. Silk fibroin can be converted into water-soluble regenerated silk fibroin protein (RSF) through a series of treatments. Under specific conditions, RSF solutions can be processed into films, sponges, microspheres, gels, and nanofibers for various applications [7], including biomedicine [8,9,10] (e.g., drug delivery carriers, wound dressings, and tissue adhesion), tissue engineering [11,12,13] (e.g., tissue scaffolds), food processing [14,15] (e.g., food additives and packaging), effluent treatment [16] (e.g., water filtration membranes), optics [17,18] (e.g., nanolithography and optical fibers), electrochemistry [19,20,21] (e.g., electrocatalytic materials, supercapacitors, and nanogenerators), and biosensing [22,23] (e.g., flexible wearable sensors and human–machine interaction).

Current evidence suggests that the molecular weight and protein structure of RSF vary depending on the extraction method used [24,25,26], consequently impacting its mechanical strength [27,28], degradability [29,30], biocompatibility [31], and thermal stability [32,33], which in turn affect the processing and application of RSF materials. The extraction process of RSF comprises four steps: degumming, dissolution, dialysis, and centrifugation [34]. Degumming involves heating raw silk in an alkaline aqueous solution to separate silk fibroin from sericin, which dissolves in water. Silk fibroin devoid of sericin remains poorly soluble in water due to the crystalline structure formed by the extensive repetitive hydrophobic motifs within the heavy chains of SF [35,36]. The dissolution step converts insoluble SF into soluble RSF through the interaction between solvent molecules and SF, facilitating subsequent processing. At present, there have been many studies on SF dissolution methods, such as LiBr [37], LiSCN [38], Ajisawa method [39], and Mu solvent [40]. Wang et al. summarized the SF fiber dissolution systems in their recent review [41]. Other reviews summarized the forming process of RSF, including compositions [42,43], nanofibers [44], and chemical modification of silk fibroin [45]. However, these excellent reviews focused on more than the extraction methods, forming process, and application rather than the relationship between the extraction methods and characters of RSF and the effect of RSF properties on the forming process and application. Actually, the characters of RSF were deeply affected by the extraction process, and the properties of RSF deeply influence the further forming process and features of RSF materials and their applications.

This review aims to elaborate and summarize various dissolution methods of SF while analyzing the effects of different SF dissolution methods on the structure and performance of RSF, as shown in Figure 1. Overall, our findings establish a systematic link between the dissolution process and the structure and characters of RSF, providing valuable references for the future forming process, research, and application of RSF in diverse fields.

## 2. Silk Fibroin

Silk protein consists of silk fibroin (70%), sericin (25%), and minor impurities (5%) [46]. The silk structure mainly comprises two silk fibroin fibers and is covered with a sericin layer containing small nonprotein impurities [47]. Sericin is a water-soluble protein with a loose and disordered spatial structure, which plays the role of lubrication, protection, and adhesion in cocoon silk [48]. Silk fibroin is a fibrous protein composed of a heavy chain (H-chain) (390 kDa), light chain (L-chain) (26 kDa), and glycoprotein P25 (30 kDa) at a molar ratio of 6:6:1 [49]. SF contains 18 kinds of amino acids, with nonpolar amino acids, such as glycine, alanine, and valine, accounting for about 70.5% and polar amino acids, such as serine and tyrosine, accounting for about 29.5% [50]. The H-chain of SF consists mainly of 12 extensive repetitive domains separated by 11 minor nonrepetitive domains [2]. The repetitive domains are composed of a high content of Gly-X (X = Ala, Ser, Thr, and Val) dipeptide repeat motif and can organize themselves into microcrystalline domains via intramolecular/intermolecular forces, such as hydrogen bonding, van der Waals forces, and hydrophobic interactions [36,51]. The nonrepetitive domains with irregular but conserved GT~GT sequences are the linkers between the microcrystalline domains [52]. The amino acid composition of the nonrepetitive domains is diverse, including glutamic acid, aspartic acid, arginine, lysine, and proline [53]. The contained proline residue can reverse the orientation of the chains of the H-chain through the formation of a five-membered ring on the backbone, which may be a major factor for antipolar β-sheet formation [52]. Furthermore, the N-terminal 151 residues and C-terminal 50 residues of the H-chain are completely nonrepetitive and amorphous [35,52]. The L-chain shows greater hydrophilicity and elasticity than the heavy chain, due to nonrepetitive amino acid sequences with low or no crystallinity [54].

In the solid state, there are two crystal forms of Silk I and Silk II in silk fibroin [55]. Silk I is mainly composed of random coil/β-turns (type II), which constitutes the amorphous or semicrystalline domain of SF, and its content is positively correlated with the elasticity of silk fibers [56]. Silk II is the main crystalline domain of SF, composed of antipolar β-sheet formed by repeated folding of (GAGAGS)n through β-angles, and its content determines the mechanical strength of the silk fibers [57]. When treated with a specific solvent, SF molecules gradually dissolve due to changes in the conformation and interaction. Different solvents can affect the conformation and molecular weight of RSF, which affect the properties of membranes, nanomaterials, and scaffolds prepared with RSF as substrates.

## 3. SF Degumming

Degumming is an integral step for RSF preparation, potentially affecting the structural integrity of RSF. Extensive methods have been reported on the process of degumming, including alkaline degumming [58,59], acid degumming [60], biological degumming [61], and physical degumming [62]. In particular, Na_2_CO_3_ is the most common degumming reagent with the advantages of simple operation and a high degumming rate [34,63,64,65]. However, the Na_2_CO_3_ solution destroys the molecular chains of fibroin proteins due to its strong alkalinity and deteriorates the mechanical properties and thermal stability of the RSF materials [66,67,68]. The excessive heating temperature and time will unavoidably increase the degradation of SF [69,70]. As a destroyer of hydrogen bonds, urea can alter the conformation of the silk sericin molecule through the breaking of hydrogen bonds, thereby promoting degumming [71,72]. At the same solubilization conditions, RSF prepared by urea degumming has a higher molecular weight, crystallinity, and solution viscosity than that prepared by Na_2_CO_3_ and citric acid degumming, which is advantageous for the preparation of high-performance RSF materials [73,74,75]. Organic acids are generally milder and less destructive to SF than alkali [76]. With the treatment of citric acid, the structure and properties of SF fibers are preserved to the maximum extent, but the degumming rate is slow [77]. As a new green degumming agent, the protease efficiently hydrolyzes specific sites of the sericin at low temperatures of 50–65 °C [78]. The ultrasonic method can disintegrate silk sericin agglomerates into small particles at 60 °C through high frequency acoustic energy. It is difficult to obtain a high concentration of RSF due to its low degumming rate [79].

Common degumming processes of silk and their impact on SF are summarized in Table 1.

## 4. SF Dissolution Processes and Their Influence on RSF Characters

### 4.1. Methods and Properties of RSF by Acid Dissolution

Acids capable of dissolving SF fibers include inorganic acids, such as phosphoric acid [81,82], sulfuric acid [83], and hydrochloric acid [84], and organic acids, like formic acid [85]. In 1989, Ishisaka successfully dissolved fibroin using concentrated phosphoric acid [81]. When SF fibers are treated with phosphoric acid, they lose most of the intermolecular bonds and transform from a stable β-sheet structure to an unstable random coil, destroying the SF crystal structure [82]. Over time, the silk peptide chain gradually undergoes hydrolysis. The viscosity of the solution decreases with an increasing degree of hydrolysis [81]. Films prepared from phosphoric acid-extracted fibroin have tunable nanostructures, excellent stretchability, outstanding biocompatibility, and good wound-healing effects, which suggest they are promising candidates for full-thickness skin defect repair [86]. Phosphoric acid also allows the preparation of highly stable Pickering emulsions with regenerated nanosilk as the emulsifier for the food and cosmetic industries [87].

Sulfuric acid can be harnessed to prepare silk fibroin peptides with good water solubility, but the hydrolysis process destroys tryptophan and partially damages serine, tyrosine, and threonine [83]. Compared with sulfuric acid alone, microwave-assisted dissolution can shorten the dissolution time by 4 h and increase the yield of silk peptide to 69% [88]. Additionally, the extent of hydrolysis affects the length of the peptide chain and the content of free amino acids [83]. Sulfuric acid can also be used to prepare nanofibers with the original crystal structure of silk by directly exfoliating degummed SF [89]. The silk nanofibers have excellent stability at either acidic (pH = 3) or alkali (pH ≥ 7) conditions, which can be fully blended with polymers in different pH environments to prepare new advanced composite materials [89]. Sulfuric acid can also be used to prepare silk protein nanowhisker suspensions, which can be used to strengthen the RSF (extracted by 9.3 M LiBr) material films [90].

Hydrochloric acid exhibits a strong hydrolysis effect on SF, resulting in a high content of oligopeptides and free amino acids in the prepared RSF solution [84]. Hydrochloric acid dissolution is often used to prepare soluble SF powder, showing great potential in the development of functional foods [91].

Formic acid, on its own, cannot disrupt the Silk II crystalline structure of natural SF for dissolution [92]. However, the molecular weight and crystallinity of RSF decrease compared with those of natural SF, enabling formic acid to independently dissolve RSF fibers [93,94]. Formic acid is commonly mixed with acid, such as phosphoric acid [85,95] and hydrochloric acid [96], or inorganic salts, such as CaCl_2_ [97], LiBr [98], CaBr_2_ [99], LiCl, and Ca(NO_3_)_2_ [100]. During the dissolution process, formic acid interacts with the polar groups in SF molecules to preserve SF integrity [101]. Concentrated formic acid enhances the solvation ability of the mixed solvent, reduces the degradation rate of SF, and improves the stability of the SF solution [102]. Considering the effect of solubility and solution viscosity, the optimal ratio of phospho–formic acid is 20/80 and 30/70 [85]. The RSF solution prepared using a phospho–formic acid mixed solvent can be directly used in wet spinning or an electrostatic spinning process without dialysis to prepare RSF fibers or scaffolds [95]. The RSF filaments maintain a β-sheet structure and exhibit excellent mechanical properties [85]. With the assistance of ultrasonic waves, degummed SF can also be dissolved in the mixture of a solution containing 30% formic acid and 0.5% hydrochloric acid to prepare powdered nanofibers [96]. Acid dissolution is frequently employed to prepare SF peptides, powder, or RSF filaments applied in spinning, food, cosmetics, biomedicine, and other fields [103].

The preparation methods and properties of RSF obtained by acid dissolution of SF are shown in Table 2.

### 4.2. Methods and Properties of RSF by Alkali Dissolution

Due to its strong alkaline properties, the NaOH solution causes the breakage of hydrogen bonds and hydrolysis of the peptide chains in the SF filamentous structure [105,106]. The resulting RSF comprises short fibrillar SF proteins with molecular weights typically less than 66.2 kDa [107]. The prepared RSF short fibers are at the micron level, and the length of the fibers is controlled by adjusting the dissolution temperature, time, and alkali concentration [108], which can be added to the materials based on RSF extracted by other systems to enhance their mechanical properties [106,109]. To facilitate the dissolution of raw silk in a single step, a certain proportion of urea can be added to the alkali solution. The formed urea hydrate hinders the reformation of hydrogen bonds between SF molecules, thereby accelerating the rate of SF fiber dissolution [110]. Prepared silk protein has a significantly high inhibition rate of tyrosinase and stronger chelation ability to ferrous ion and moisture retention ability, exhibiting skin care characteristics of whitening and antioxidation [110]. Consequently, the alkali dissolution of SF finds applications in fields such as cosmetics [111], hemostatic materials [108], and tissue engineering [105,106,109].

### 4.3. Preparation Methods and Properties of RSF by Salt and Salt Complex System Dissolution

#### 4.3.1. Salt

High concentrations of neutral salts, such as LiBr [112,113,114], LiSCN [115,116], CaCl_2_ [117,118], and Ca(NO_3_)_2_ [119], under hydrothermal conditions, can dissolve SF. The presence of lithium salts in the solution disrupts the intermolecular hydrogen bonds and van der Waals forces within SF due to the strong polarity of the ions, promoting the dissolution of SF [120,121]. It is widely recognized that chains of SF molecules dissolved by LiBr solution are prone to free extension and aggregation depending on hydrophobic/hydrophilic interactions at high shear stresses, resulting in the transition from shear thinning to shear thickening with an increasing shear rate [122]. At high concentrations, LiBr efficiently dissolves SF, resulting in a highly concentrated RSF solution prone to gelation [123,124]. However, LiBr degrades the multilevel structure of silk protein, leading to a decrease in molecular weight, which hinders the transformation into a β-folded structure when forming films [125]. Therefore, the RSF materials from LiBr contain more random coil and β-turns compared with those from other systems and are suitable for studying SF conformational changes. But the ductility and stability of RSF materials decreased due to lower crystallinity [124]. The electrospinnability of RSF was slightly deteriorated with the increase in extraction temperature [114].

LiSCN, on the other hand, can directly dissolve raw silk fibers at room temperature with minor degradation of SF without degumming [115]. The ability to ensure the integrity of SF makes it suitable for studying the composition and structural changes of silk fibroin [126,127]. The intact fibroin solution also shows a strong tendency to form a gel [115]. The films dried directly from the RSF solution are mainly amorphous [116]. With the treatment of acidic reagents (e.g., formic acid and trifluoroacetic acid) or ethanol, the secondary structure of RSF films is altered to modulate their biodegradability and mechanical properties for cell scaffold materials [128]. In fact, LiSCN is more commonly employed to dissolve Antheraea pernyi silk fibroin [129,130,131]. The RSF of Antheraea pernyi obtained by alkaline degumming and LiSCN solubilization showed considerable degradation, with dispersed molecular weight bands from 20 kDa to 150 kDa [130]. However, thiocyanates are toxic and hazardous to humans, limiting their application in food additives, cosmetics, and biomedicine [132].

In a calcium salt solution, SF fibers swell, and Ca^2+^ ions penetrate SF, complexing with tyrosine and serine side-chain groups [124]. This causes a change in the SF molecular structure from β-folded to a random coil/α-helical conformation [133]. High concentrations of CaCl_2_ promote the formation of the hydrated layer on the surface of SF due to the production of strong polar ions, increasing the hydrophilicity of SF and weakening the intermolecular interaction forces [134]. As a result, the dissolution rate of the SF filament is improved with increasing calcium salt concentrations [134]. However, if the concentration of Ca^2+^ is too high, it will then seize the water molecules bound to the protein surface, destroying the hydration layer around the protein, thus causing a decrease in the dissolution rate [134]. Compared with the CaCl_2_–ethanol–water system, the RSF solution prepared by CaCl_2_ systems has a higher gelation rate, and the formed gel contains more β-sheet with a more developed three-dimensional network [117]. Ca(NO_3_)_2_ has a relatively inferior solubilization effect on silk fibers, especially at low temperatures or concentrations [119]. RSF solutions prepared using calcium salts tend to have higher viscosity and turbidity [119]. However, the small amount of residual Ca^2+^ has minimal effects on subsequent material processing and conformational changes [124].

The preparation method and properties of RSF obtained by salt dissolution of SF are summarized in Table 3.

#### 4.3.2. Salt–Acid Complex System

Formic acid can serve as a cosolvent to promote the swelling of SF fibers, aiding in the penetration of metal ions from the salt solution into SF fibers [142]. This process breaks the intermolecular hydrogen bonds of SF, thereby enhancing the efficiency of SF dissolution [142]. While SF solutions prepared using salt systems are prone to gelation, SF remains highly stable in salt–formic acid solutions, preventing aggregation, precipitation, and gelation [101]. The molecular weight of RSF extracted by the salt–acid system has not been reported for the time being, probably because very little attention has been paid to it. RSF prepared using salt–acid systems exhibits a nanoscale fiber structure with improved mechanical properties, enhanced moisture permeability, and excellent biocompatibility and biodegradability [143]. This makes it suitable for applications such as electrostatic or fiber-spinning materials [144,145], biomedicine [146,147], and stent materials [148].

CaCl_2_–FA dissolves interfibrillar parts consisting mainly of amorphous regions while preserving the nanofiber structure, promoting the preparation of regenerated filamentous materials with higher strength and ductility [148,149,150]. The high concentration of CaCl_2_ converts the partial β-sheet into noncrystalline structures, such as α-helices and β-turns, by breaking the intramolecular hydrogen bonds in SF, increasing the structural disorder degree. Therefore, with an increasing CaCl_2_ concentration from 1.0% to 6.0%, the morphologically of RSF turns from microfibrils to nanoparticles, and the ductility of those RSF materials increases, but the stiffness, viscoelasticity, and thermal stability decrease [151,152]. RSF filaments have a typical β-sheet structure and exhibit excellent enzymatic degradation properties, biocompatibility, and outstanding strength (breaking stress of 276.4 MPa) and ductility (elongation at break of 40.8%) after stretching [97].

In the effect of LiBr–FA, SF lost numerous intermolecular hydrogen bonds and was decomposed into amorphous nanofibers with diameters of 10–20 nm and lengths of 200–350 nm [153]. The SF nanofibril structure enables RSF solutions to exhibit excellent spinnability in electrostatic spinning and significantly improves the mechanical properties of RSF materials. In addition, the average diameter of SF nanofibers in the solution can be controlled by adjusting the concentration of the LiBr–FA solution. This feature facilitates the preparation of scaffold materials with various hierarchical micronanofibrous structures that support cell proliferation and adhesion [154]. The RSF film prepared is a network structure dominated by intramolecular β-sheets between Silk I and Silk II, contributing to water insolubility and good ductility [155]. RSF can be stored at room temperature for more than 3 months in the form of freeze-dried powder, exhibiting adequate stability [153].

Among the three salt–acid systems, CaBr_2_–FA demonstrates an optimal effect on filament solubilization. And the SF nanofibers with small diameters and short lengths tend to cluster together during fabrication, resulting in a dense surface of the RSF film prepared by them [99]. The CaCl_2_–FA [149,150,151] and LiBr–FA [153,154] systems are widely used in silk fibroin dissolution of *Bombyx mori*, while CaBr_2_–FA [99] has been less-studied and less-applied. At present, LiCl–FA and Ca(NO_3_)_2_–FA have only been studied for the silk fibroin dissolution of Argema mimosae [100].

The methods of preparing RSF by dissolving SF in the salt–acid system and the properties of the obtained RSF are summarized in Table 4.

#### 4.3.3. Salt–Alcohol–Water System

The salt can also be mixed with alcohol and water in a specific ratio to form a ternary solution for SF dissolution. In the salt–alcohol–water ternary system, water acts as a swelling agent, expanding the noncrystalline regions of SF [160]. Alcohol molecules can enter the crystallization regions of the filamentous protein chains, reducing the surface tension of SF and weakening the hydrophobic interactions [39,124]. This increases the permeability of hydrated metal ions and accelerates the decomposition of SF chains. Compared with LiBr, the CaCl_2_–ethanol–water ternary system has become one of the most commonly used methods for SF dissolution due to its low cost [161], high productivity [124], harmlessness to organisms [162], and environmental friendliness [163]. In a study by Cho et al. [112], the molecular weight (Mw) bands of SF were determined using fast protein liquid chromatography. It was observed that the Mw bands of 450 kDa decreased, while 150 kDa and 16 kDa increased with solubilization time in the CaCl_2_–ethanol–water ternary system, suggesting that the system may selectively cleave the SF molecular chains at specific sites. The SF crystalline structure is destroyed upon dissolution, resulting in broken lamellae from ribbons. After drying, the RSF appears as an irregular granular powder [164]. The degradation degree of SF molecular chains dissolved by CaCl_2_–ethanol–water is higher than by LiBr systems [24]. It is worth mentioning that there is no significant difference in the biocompatibility of RSF materials prepared by CaCl_2_–ethanol–water and LiBr systems [163]. Compared with other ternary reagents, the film from CaCl_2_–ethanol–water maintains more Silk I, probably as a result of its superior ability to protect the conformational and intermolecular arrangement of RSF [165]. Analyzing the IR spectrogram of RSF solutions, Lingling Li et al. [166] found that the RSF solution dissolved in the CaCl_2_–methanol–water system exhibited more β-structure and less decrystallization compared with the RSF solution dissolved in the CaCl_2_–ethanol–water system. This may indicate a weaker interaction of methanol with SF chains than ethanol.

In contrast to the CaCl_2_ ternary system, the RSF obtained from the Ca(NO_3_)_2_ ternary system has a smaller molecular weight and undergoes more extensive filament degradation [165]. In the Ca(NO_3_)_2_–methanol–water system, RSF exhibits a relatively uniform granular shape. However, decreased crystallinity and the smaller molecular weight reduce the thermal stability and mechanical property of the RSF materials [167]. However, the poor mechanical properties of RSF materials can be improved through using acidic spin systems and stretching SF fiber during processing [168]. The Ca(NO_3_)_2_–ethanol–water system poorly dissolves SF, producing a small amount of undissolved silk fibers [166]. The RSF materials treated with these calcium–alcohol solvents are water-soluble [165]. The LiBr–ethanol–water system demonstrates a rapid dissolution rate and high yield. However, the viscoelasticity of the obtained solution is proportional to the concentration of RSF and the ratio of ethanol to water in the solvent [169].

The rheological behavior of the SF solution can reflect the solvation in the dissolved system [122]. Solvation primarily involves electrostatic interactions between solvent and SF molecules, the intensity of which affects the conformational stability of SF molecules to a certain extent [170]. SF molecules have a unique amphiphilic block structure that gives them an inherent tendency to form β-sheet under favorable conditions [35,171]. Due to the inherent tendency to form β-sheet and the weak solvation of LiBr, SF molecular chains tend to extend freely to form intermolecular interactions and aggregate with an increasing shear rate, resulting in shear thinning followed by shear thickening behavior [122]. The strong solvation in the Ca(NO_3_)_2_–methanol–water, LiBr–ethanol–water, and CaCl_2_–ethanol–water systems allows SF solutions to maintain near-constant viscosity at high shear rates [122]. The hydrophilic/hydrophobic interactions of the SF chains make them prone to self-assembly in water, forming micelle structures [172]. The average diameters of RSF micelles prepared from different systems are as follows: CaCl_2_ (128.8 nm) > LiBr (82.7 nm) > CaCl_2_–ethanol–water (63.2 nm) > LiBr–ethanol–water (60.5 nm) > Ca(NO_3_)_2_–methanol–water (33.89 nm). Smaller micelle diameters indicate a higher degree of SF degradation by the solvent [124]. Among the above systems, the Ca(NO_3_)_2_–methanol–water system is the most destructive for SF.

The gelation rate of RSF solution is influenced by the SF concentration, temperature [173], and molecular weight [174]. Increasing the SF concentration and temperature accelerate the gelation rate of the RSF solution and result in narrower pore sizes in the prepared hydrogel. Smaller pore sizes allow for more even stress distribution, leading to higher compressive strength and modulus [173]. The molecular weight is directly proportional to the gelation rate of the RSF solution and inversely proportional to solution stability [174,175]. The molecular weight is closely related to the properties of the RSF solution and the resulting material. A high-molecular-weight RSF solution exhibits high viscosity and a fast gelation rate, resulting in a densely packed spatial structure and excellent mechanical properties in the prepared RSF material. On the other hand, low-molecular-weight RSF forms nanoparticles with smaller diameters, superior solution stability, less tendency to gel, and excellent degradability in the resulting RSF material [145,176]. As shown in Table 4, the gel properties of RSF solutions prepared by different systems can be roughly evaluated in terms of the summarized molecular weights.

The methods of preparing RSF by dissolving SF in the salt–alcohol–water system and the properties of the obtained RSF are summarized in Table 5.

### 4.4. Preparation Methods and Properties of RSF by Ionic Liquid Dissolution

Ionic liquids are ionic systems composed of organic cations and anions (organic or inorganic) in a liquid state slightly above room temperature [183]. They are new, nontoxic, and environmentally friendly solvents with high solubility for many organic, inorganic, and metal–organic compounds and polymer materials [184]. The solubility of SF in ionic liquids depends on the nature of the cations and anions, with the anions having a more significant effect [185]. During the dissolution process, organic cations, such as imidazole and TBA^+^ liquids, can partially weaken the hydrophobic effect through steric hindrance or electrostatic repulsion [186]. Anions, such as Cl^−^ and OH^−^, interact with hydroxyl protons and amino groups on the SF chain to form hydrogen bonds, which weaken or further disrupt the original SF hydrogen bonding network from both the outside and inside, thereby promoting dissolution [185].

Due to the interactions between ionic liquids and SF molecules, SF has good solubility in ionic liquids [187]. And the obtained RSF solution mixed with ionic liquid has a high stability and a slow gelation rate, allowing it to be stored for a long time [188,189]. Slowly adding a certain proportion of water to the mixed solution of SF and ionic liquid can reduce the viscosity of the solution without precipitation [190,191]. However, the degree of SF solvation decreases with increasing water content [189]. When the water content exceeds a certain ratio, ionic liquid/H_2_O becomes a poor solvent to RSF, and the hydrophobic moieties of SF reform a β-sheet through hydrophobic solvent action and intra/intermolecular hydrogen bonding, which in turn facilitates the sol-to-gel transition [189]. In addition, along with the addition of polar organic solvents called coagulants, such as methanol or ethanol, to the mixed solution, RSF can be quickly separated from ionic liquids in the form of a precipitate [192]. The polar organic solvent can be further removed by volatilization to prepare pure RSF nanoparticles. This system avoids long dialysis and concentration steps and greatly saves time [192]. It is also environmentally friendly because ionic liquids can be recycled through rotary evaporation [187].

Ionic liquids containing Cl^−^ ions, such as 1-allyl-3-methylchloroimidazole ([AMIM]Cl), exhibit an increasing solubility with increasing temperature, reaching 14.5% at 100 °C [193]. Vortex stirring or ultrasonic treatment is commonly used to assist in the dissolution process [194,195]. The sealed SF/AmimCl solution can be stored at room temperature for more than 1.5 years due to the stability of ionic liquids [196]. The protein in low-concentration SF/[AMIM]Cl solutions exists mainly as dispersed individual chains [196]. When the concentration of SF is increased to more than 3%, SF molecular chains tend to overlap and associate with each other, which makes the whole solution like Newtonian fluid in the low-shear-rate region, while at high shear rates, macromolecules undergo a process of rearrangement to reduce intermolecular friction, resulting in shear thinning [196]. As coagulants, ethanol and n-butanol can facilitate the transformation of SF conformation from the random coil to the β-sheet during SF regeneration. And the RSF film exhibits a good wet-state mechanical strength and a small water solubility loss rate due to their mainly β-sheet structure [193].

However, excessive temperatures can lead to increased SF degradation [197]. To address this, researchers have developed alkali-based aqueous solutions of ionic liquids, such as choline hydroxide aqueous solution and tetra-butyl ammonium hydroxide. These solutions can dissolve SF under mild conditions and preserve the structural integrity of SF. In the study by Muhammad on the choline hydroxide aqueous system [198], the solubility of SF reaches up to 25% at 50 °C. Regenerated nanoparticles with an average particle size of 134.4–395.2 nm can be obtained by methanol treatment. And RSF nanoparticles have a negatively charged surface, preventing further agglomeration of the particles by electrostatic repulsion, so that the nanoparticles can remain stable for a long time in the cell culture medium [198,199]. In contrast, SF rapidly dissolves into a tetra-butyl ammonium hydroxide solution at room temperature. The higher the concentration of SF, the cloudier the solution, which is probably attributed to the formation of RSF-transient aggregates in the solution [200]. This dissolution system has been used to prepare a novel multilayered silk nanofibril-based membrane for water filtration systems [201].

The preparation of RSF by ionic liquid dissolution of SF and the properties of the obtained RSF are summarized in Table 6.

### 4.5. Preparation Methods and Properties of RSF by Enzyme Dissolution

SF is composed of 18 amino acids, with a high content of glycine and alanine [207]. When SF is dissolved by acid [83] or alkali reagents [106], some amino acids, such as tryptophan, serine, and arginine residues of SF, are partially destroyed or racemized, reducing the glossiness of the RSF fibers. Enzymatic reactions used for SF dissolution have mild reaction conditions and are less destructive to the composition and structure of SF [208]. Moreover, enzymes exhibit specificity, making it easy to control the degree of SF hydrolysis [209]. Commonly used proteases for SF dissolution include alkaline protease [210,211,212], papain [213], and pancreatic protease [214]. Alkaline protease and papain have broad specificity and can act on a wide range of peptide bonds, degrading the crystalline region primarily composed of glycine and alanine, thus demonstrating excellent SF hydrolysis activity [215,216]. On the other hand, trypsin specifically degrades peptide bonds formed by lysine (Lys) and arginine (Arg), selectively targeting the noncrystalline region of SF. This can result in more precipitates within the solution, indicating a narrower range of action [217].

The silk peptide prepared by alkaline protease has a particle size of 100–500 nm and a lamellar morphology due to agglomeration of RSF [210]. After freeze-drying, the RSF peptide powder shows excellent scavenging activity of 2,2-diphenyl-1-picrylhydrazyl (DPPH) radical, demonstrating significant antioxidant properties [211], while the composites based on chitosan and modified by silk peptides exhibit excellent moisture absorption, retention properties, and pronounced cell compatibility for wound dressing applications [212]. The silk peptide treated with papain possesses anti-inflammatory activity by suppressing the activation of ERK, which can be applied as food ingredients and skincare products [213]. Trypsin-hydrolyzed SF peptides are nanoscale and have been applied in the preparation of nanometer silk fibroin peptide/polymer composite biomaterials [214].

The products obtained from SF enzymatic hydrolysis are mixtures of amino acids and oligopeptides [218]. These products have small molecular weights and can be easily absorbed by the human body [107]. They exhibit various functions, such as hypoglycemic [218], hypocholesterolemic [219], and antioxidant [211]. Due to these properties, they hold significant application value in fields like functional health food [220], cosmetics [213], and medical materials [221].

The preparation of RSF by enzyme dissolution of SF and the properties of the obtained RSF are summarized in Table 7.

## 5. Perspectives

Silk protein has been extensively applied in biomedicine, biosensing, and wearable equipment due to its excellent biocompatibility, chemical modifiability, and mechanical properties. However, due to internal hydrophilic, hydrophobic, and hydrogen bonds, silk protein is challenging to dissolve in conventional solvents. Various solubilization systems have been developed to extract natural silk protein, including acid, base, salt, salt–acid, salt–alcohol, ionic liquid, and enzyme systems. However, there is still a lack of systematic and comprehensive studies on the properties of regenerated silk fibroin (RSF) prepared using each solubilization system. The influence of various factors within each system on the properties of RSF has not been fully explored, and comparative studies between different systems are also lacking. Furthermore, while there are numerous methods available for preparing RSF, LiBr dissolution is commonly used, and the resulting RSF is often prepared in the form of membranes, hydrogels, porous scaffolds, and other structures for applications, such as tissue scaffolds, wearable devices, and sensors. In future research, selecting a more suitable extraction system based on the performance of the RSF obtained using different extraction methods would be beneficial. This approach would help identify the optimal preparation scheme and achieve optimal performance for specific applications.

## Figures and Tables

**Figure 1 ijms-24-13153-f001:**
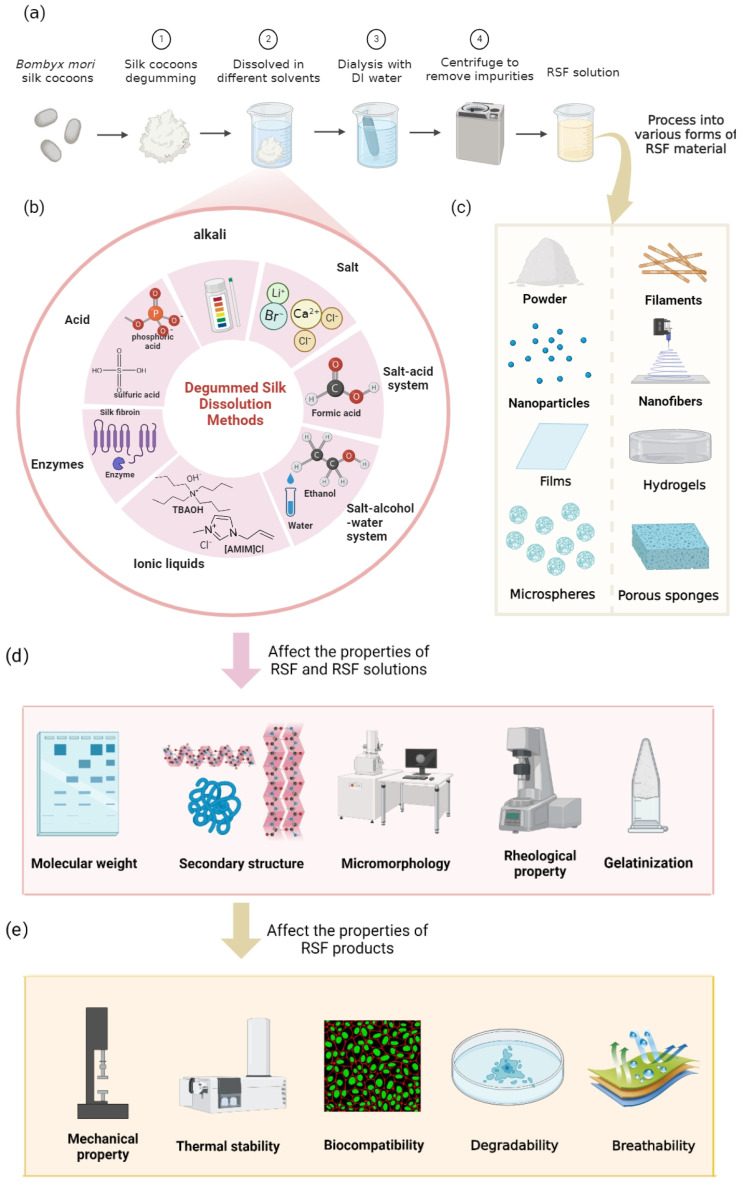
Schematic showing the preparation and subsequent processing of regenerated silk fibroin (RSF): (**a**) The preparation process of RSF comprises four steps: degumming, dissolution, dialysis, and centrifugation. (**b**) Different dissolution systems for degummed silk. (**c**) Under specific conditions, the prepared RSF solutions can be processed into various forms of materials. (**d**) The properties of RSF and RSF solutions, including molecular weight, secondary structure, micromorphology, rheological properties, and gelatinization. (**e**) The properties of RSF products, including mechanical properties, thermal stability, biocompatibility, degradability, and breathability.

**Table 1 ijms-24-13153-t001:** The common degumming processes of silk and the effects on SF.

Degumming Reagents/Types	Concentration	Temperature, °C	Time, min	Cocoon-to-Liquor Ratio	Effects on SF	References
Na_2_CO_3_	1 g/L Na_2_CO_3_	90	60	1:40	SF fibers display a clean fiber surface with only a few deposits and a light coating of the residual sericin.	[62]
	0.2 mol/L Na_2_CO_3_	90	30	-	The light and heavy chains of SF are slightly degraded, and the structure is relatively complete.	[70]
	0.5 wt% Na_2_CO_3_	100	30	1:40	Various rod-like deposits instead of individual fibrils are visible on the surface of the degummed SF.	[66]
Urea	8 mol/L Urea	90	180	1:30	SF fibers have an average particle diameter of 221.1 nm with little or no silk sericin residue on the surface.	[75]
Citric acid	15 wt% Citric acid	98	30	1:20	SF fiber surface is highly smooth, showing very fine longitudinal striation. The molecular conformation does not change and shows β-sheet and random coil.	[77]
Serine protease	crude enzyme	50	120	-	SF fibers are bleached, and the surface is extremely smooth without silk sericin residue.	[80]
Ultrasonic treatment	-	60	30	1:200	SF fibers have sericin residue, and mechanical strength decreases.	[65]

**Table 2 ijms-24-13153-t002:** Preparation of RSF by acid dissolution of SF and properties of the obtained RSF.

Solvent	H_3_PO_4_	H_2_SO_4_	HCl	H_3_PO_4_–HCOOH
Dissolution Method	11.5 wt% degummed SF dissolves in 85% H_3_PO_4_ solution at room temperature for 30 min [81]	Degummed SF dissolves in a 20 wt% H_2_SO_4_ acid solution at 120 °C for 6 h [83]	Degummed SF dissolves in 4 mol/L HCl solution at 98 °C in a water bath for 40 h [84]	Degummed SF dissolves in H_3_PO_4_–HCOOH solution at room temperature for 2 h [85]
Color	Light brown	Dark brown	Yellow	Light brown
Molecular weight	-	25–100 kDa	10–100 kDa	-
Secondary structure	Mainly random coil with small amount of β-sheet	Mainly random coil	Mainly random coil/β-sheet	Mainly random coil
Rheological properties	Shear thinning	-	-	Shear thinning
Gelation	E ^1^	E	E	E
Film-forming properties	G ^2^	-	-	G
RSF material properties	RSF films have excellent tensile strength and ductility with 143% breaking strain	Silk peptides are very water-soluble and are easily absorbed by digestive organs and skin.	Silk powder has good water solubility and poor thermal stability with a degradation temperature of 241.5 °C.	RSF filament has excellent mechanical properties with the tenacity and breaking strain of 2.3 gf/d and 18%.
Processing forms	Filament, porous film, peptide	Natural amphoteric nanofiber, peptide	Powder, peptide	Filament, film, powder, peptide
Applications	Wound healing [86], spinning [81], food, cosmetic [87], biosensing [104], waste cocoon recycling [88]	Biomedicine [89], food processing [84]	Food processing [91]	Tissue engineering [95], spinning [85]

^1^ SF solution is easy to gelate (E); ^2^ SF solution has good film-forming properties (G).

**Table 3 ijms-24-13153-t003:** Preparation of RSF by salt dissolution of SF and properties of the obtained RSF.

Solvent	LiBr	LiSCN	CaCl_2_	Ca(NO_3_)_2_
Dissolution Method	2 wt% degummed SF dissolves in 9 mol/L LiBr solution at 80 °C [122,124]	Degummed SF dissolves in 9 mol/L saturated LiSCN solution at 30 °C for 35 min [135]	2 wt% degummed SF dissolves in 50 wt% CaCl_2_ solution at 80 °C for 4 h under vigorous magnetic stirring [124]	2 wt% degummed SF dissolves in 50 wt% Ca(NO_3_)_2_ solution of at 70 °C for 6 h [119]
Color	Light yellow	Slightly milky	White	Light yellow
Molecular weight	25 kDa and 60–100 kDa	20 kDa or 30 kDa–200 kDa or more	25 kDa and 66.2–100 kDa	-
Secondary structure	Mainly random coil with small amount of β-sheet	Random coil	Random coil	Random coil
Rheological properties	Shear thinning at low shear rate, shear thickening at high shear rate	-	-	-
Gelation	E ^1^	E	E	E
Film-forming properties	G ^2^	G	G	G
RSF material properties	RSF film has poor stability with thermal degradation temperature of 278 °C and water solubility. RSF fibers have good electrospinnability, outstanding stiffness with tensile strength of 210 MPa, and poor ductility with elongation at break of 11%.	RSF films are mainly amorphous and have high thermal stability with a thermal decomposition temperature of 286 °C.	RSF film conformation is between the structure of Silk I and Silk II, with a thermal decomposition temperature of 288 °C.	RSF film is Silk I crystalline conformation.
Processing forms	Film, gel, fiber, nanoparticle	Film, powder	Filament, film, powder, gel	Film
Applications	Drug delivery [136], tissue adhesion [137], tissue engineering [138], food packaging [139], electrochemistry [140], biosensing [141]	Biomedicine, tissue engineering [128], protein analysis [126]	Biomaterials, waste cocoon recycling [134]	Biomedicine [119]

^1^ SF solution is easy to gelate (E); ^2^ SF solution has good film-forming properties (G).

**Table 4 ijms-24-13153-t004:** Preparation of RSF by salt–acid system dissolution of SF and properties of the obtained RSF.

Solvent	CaCl_2_–HCOOH	CaBr_2_–HCOOH	LiBr–HCOOH
Dissolution Method	Degummed SF dissolves in 4% (*w*/*v*) CaCl_2_–HCOOH solution with stirring at room temperature for 4 h [99,154]	Degummed SF dissolves in 4% (*w*/*v*) CaBr_2_–FA solution with stirring at room temperature for 2 h [99]	Degummed silk dissolves in 2% (*w*/*v*) LiBr–FA at room temperature for 3 h [145,154]
Color	Light yellow	Dark yellow	Yellow
Molecular weight	-	-	-
Secondary structure	Mainly random coil	Random coil	Random coil/α-helix
Rheological properties	First shear thickening, then shear thinning	Shear thinning	Shear thinning
Gelation	D ^1^	D	D
Film-forming properties	G ^2^	G	G
RSF material properties	The filament film is a β-sheet structure with excellent strength, ductility, biocompatibility, and biodegradability. Electrospun nanofibers have good degradability.	SF film surface is smooth and dense.	The silk fibroin film has a dense surface and β-sheet structure, and its modulus, strength, and ductility are significantly improved.
Processing forms	Filament, film, gel, nanofiber	Film	Film, nanofiber, electrostatic spinning fiber mat, SF coating
Applications	Tissue engineering [148], wound healing [146], drug delivery [147], tissue adhesion [156], electrochemistry [157], spinning [144]	-	Tissue engineering [158], drug release [159], spinning [145]

^1^ SF solution is difficult to gelate (D); ^2^ SF solution has good film-forming properties (G).

**Table 5 ijms-24-13153-t005:** Preparation of RSF by salt–alcohol–water system dissolution of SF and properties of the obtained RSF.

**Solvent**	**CaCl_2_–EtOH ^1^–H_2_O**			**CaCl_2_–MeOH ^2^–H_2_O**	
Dissolution Method	5 wt% degummed SF dissolves in CaCl_2_–EtOH–H_2_O (1:2:8 molar ratio) solution at 58 °C for 2 h [166]	5 wt% degummed SF dissolves in CaCl_2_–EtOH–H_2_O (1:2:8 molar ratio) solution at 65 °C in water bath for 1 h [165]	2 wt% degummed SF dissolves in CaCl_2_–EtOH–H_2_O (1:2:8 molar ratio) solution at 80 °C in water bath [124]	5 wt% degummed SF dissolves in CaCl_2_–MeOH–H_2_O (1:2:8 molar ratio) solution in a water bath at 58 °C for 2 h [166]	5 wt% degummed SF dissolves in CaCl_2_–MeOH–H_2_O (1:2:8 molar ratio) solution in a water bath at 65 °C for 1 h [165]
Color	Yellow	-	White and highly opaque	Yellow	Yellow and opaque
Molecular weight	25 kDa–200 kDa or more	100–300 kDa	25 kDa and 60–100 kDa	25 kDa–200 kDa or more	140–200 kDa
Secondary structure	Between random coil and β-sheet	More α-helix, type II β-turns, a few β-sheets	Mainly random coil with a small portion of β-sheet	Between random coil and β-sheet	Mainly β-sheet with partial random coil
Rheological properties	Shear thinning at low shear rates and Newtonian fluid at high shear rates			-	
Gelation	E ^3^			E	
Film-forming properties	G ^4^			G	
RSF material properties	RSF film has strong toughness with a fracture strain of 215.1% and high thermal stability with a thermal decomposition temperature of 284 °C.			Freeze-dried SF powder contains more Silk II structures versus fewer Silk I structures.	
Processing forms	Film, gel, nanoparticle, nanofiber mesh, microsphere			Film, fiber, hydrogel, nanoparticle	
Applications	Drug delivery [177], wound dressings [178], tissue engineering [179], optics [180]			Biomaterials [166]	
**Solvent**	**Ca(NO_3_)_2_–EtOH–H_2_O**		**Ca(NO_3_)_2_–MeOH–H_2_O**	**LiBr–EtOH–H_2_O**	
Dissolution Method	Degummed SF dissolves in Ca(NO_3_)_2_–4H_2_O:EtOH (1:2 molar ratio) solution in water bath at 68 °C for 2 h [166]	5 wt% degummed SF dissolves in Ca(NO_3_)_2_–4H_2_O:EtOH (1:3 molar ratio) solution with stirring at 80 °C for 30 min [124]	5 wt% degummed SF dissolves in Ca(NO_3_)_2_–4H_2_O:ethanol (1:2 molar ratio) solution at 65 °C in water bath for 1 h [165,168]	2 wt% degummed SF dissolves in LiBr–EtOH–H_2_O (45:44:11 weight ratio) solution at 80 °C [122,124]	
Color	Yellow and opaque	Yellow	Yellow	Light yellow	
Molecular weight	25–150 kDa	25 kDa and 100 kDa	95 kDa–170 kDa or more	25 kDa and 50–110 kDa	
Secondary structure	Between random coil and β-sheet	Mainly β-sheet	Mainly random coil with a small portion of ordered structure	Mainly random coil with a small portion of β-sheet	
Rheological properties	-		Linear shear thinning at low shear rates and Newtonian fluid at high shear rates	Rapid shear thinning at low shear rates and Newtonian fluid at high shear rates	
Gelation	E		E	E	
Film-forming properties	G		G	G	
RSF material properties	RSF film is a Silk II structure and has excellent thermal stability with a thermal degradation temperature of 288 °C.		RSF fiber conformation is random coil and β-sheet, with good elasticity but reduced strength.	RSF film is a Silk II structure and has good thermal stability with a thermal degradation temperature of 288 °C.	
Processing forms	Film, fiber, hydrogel, nanoparticle		Film, nanofiber, powder	Hydrogel, porous sponge, nanoparticle	
Applications	Biomaterials [124]		Drug delivery [181], spinning [168]	Biomaterials, spinning [182]	

^1^ Ethanol (EtOH); ^2^ Methanol (MeOH); ^3^ SF solution is easy to gelate (E); ^4^ SF solution has good film-forming properties (G).

**Table 6 ijms-24-13153-t006:** Preparation of RSF by ionic liquid dissolution of SF and properties of the obtained RSF.

Solvent	[AMIM]Cl ^1^	46 wt% Choline Hydroxide Aqueous Solution	TBAOH ^2^
Dissolution Method	90 °C oil bath 1.5 h [196]	Stir in 46 wt% choline hydroxide aqueous solution at 200 rpm for 2 h at 50 °C [198]	Stir in 40 wt% TBAOH aqueous solution at room temperature for 30 min [200]
Color	Amber	-	Light yellow
Molecular weight	144 kDa	-	321 kDa
Secondary structure	Random coil	random coil/β-sheet	Nearly random coil with little ordered structure
Rheological properties	Newtonian flow at a low shear rate and shear thinning at a high shear rate	-	Shear thinning
Gelation	D ^3^	D	D
Film-forming properties	G ^4^	-	-
RSF material properties	RSF film is β-sheet with a thermal degradation temperature of 310 °C.	RSF powder crystallinity decreases, and the thermal degradation temperature is 320 °C.	-
Processing forms	Powder, film, sponge, fiber	Nanoparticle, sponge, fiber, gel	-
Applications	Drug delivery [202],tissue bone engineering [203], electrochemistry [204], spinning [205,206]	Tissue engineering [198]	Effluent treatment [201]

^1^ 1-allyl-3-methylchloroimidazole ([AMIM]Cl); ^2^ Tetra-butyl ammonium hydroxide (TBAOH); ^3^ SF solution is difficult to gelate (D); ^4^ SF solution has good film-forming properties (G).

**Table 7 ijms-24-13153-t007:** Preparation of RSF by enzyme dissolution of SF and properties of the obtained RSF.

Solvent	Alkaline Protease	Papain	Trypsin
Dissolution Method	4% SF concentration, enzyme to substrate ratio 2%, and reaction at 55 °C, pH 8 for 6 h [211]	5% SF concentration, enzyme amount of 1000 U/g, and reaction at 50 °C, pH 7.5 for 240 min [222,223]	5% SF concentration, enzyme amount of 1000 U/g, and reaction at 37 °C, pH 8 for 240 min [223]
Molecular weight	5–14 kDa	14.1–20.1 kDa	<20 kDa
Secondary structure	Random coil/β-turns (type II)	Random coil/α-helix/β-sheet	Random coil/α-helix/β-sheet
Gelation	D ^1^	D	D
Film-forming properties	P ^2^	N ^3^	N
RSF material properties	RSF powder has excellent water solubility and significant procoagulant properties in vitro.	-	-
Processing forms	Film, powder, silk peptide	Powder, silk peptide	Powder, silk peptide
Applications	Wound dressings [212,221], skin care [211], food processing [224]	Food materials, skin care [213]	Food processing, biomedicine [214]

^1^ SF solutions are difficult to gelate (D); ^2^ SF solution has poor film-forming properties (P); ^3^ SF solution cannot form film (N).

## Data Availability

Data sharing not applicable.

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
