# Peer review of "Advances in Preparation and Properties of Regenerated Silk Fibroin"

_ijms, 2023, doi:10.3390/ijms241713153_

Round 1

Reviewer 1 Report

The present study comprehensively investigated the influence of various reagents, including acids, alkalis, salts, and ionic liquids, on the structure and performance of RSF obtained through their dissolution. The findings from this research provide valuable insights for researchers who are considering different dissolution methods to obtain RSF.

Generally the paper was written properly. I would like to recommend this paper for minor revision, and I suggest adding the following content:

It would be beneficial to include additional information in Table 3, specifically regarding the molecular weight of RSF produced through the salt-acid system.

Furthermore, if there is information available regarding the application of RSF in each reference cited in the table, it would be beneficial to include it alongside the corresponding references. This additional information could indicate whether RSF was applied in cosmetics manufacturing, food processing, or used as a biomaterial (with specific target tissues included for tissue engineering applications). Creating a new column in the table to present this application information would be highly advantageous.

By including the molecular weight data and information on the diverse applications of RSF cited in the references, this enhanced table will provide researchers with a more comprehensive understanding of RSF's versatile applications in various fields.

Author Response

Dear Reviewer,

Thank you for your review and comments concerning our manuscript entitled “Advances in Preparation and Properties of Regenerated Silk Fibroin” (Manuscript ID: ijms-2529320). These comments are very valuable and helpful for improving our manuscript. 

Based on the suggestions presented in your review report, we have carefully revised our manuscript. The relevant revisions in the text are shown with red highlight. We have highlighted the revisions with three colors to salute the three reviewers. At the end of this letter, the point-to-point response to the comments is attached.

Thank you again for your positive comments and valuable suggestions to improve the quality of our manuscript. We look forward to your favorable decision.

Yours sincerely,

Qing Zhang

Response of reviews comment

Comments:

The present study comprehensively investigated the influence of various reagents, including acids, alkalis, salts, and ionic liquids, on the structure and performance of RSF obtained through their dissolution. The findings from this research provide valuable insights for researchers who are considering different dissolution methods to obtain RSF.

Generally the paper was written properly. I would like to recommend this paper for minor revision, and I suggest adding the following content.

Point 1: It would be beneficial to include additional information in Table 3, specifically regarding the molecular weight of RSF produced through the salt-acid system.

Response 1: We are grateful for the suggestion. We again retrieved and read carefully the literatures about salt-acid systems, and did not find any report about the molecular weight of RSF retracted by salt- acid. Researchers possibly ignore the molecular weight of RSF. In our experience, the RSF solutions are easy to gelation in dialysis step if SF is extracted by CaCl2- methanoic acid and LiBr- methanoic acid systerms. That may be a reason the lackless of molecular weight data of RSF extracted by salt-acid system. We have added application information and molecular weights column to refine the tables.

Point 2: Furthermore, if there is information available regarding the application of RSF in each reference cited in the table, it would be beneficial to include it alongside the corresponding references. This additional information could indicate whether RSF was applied in cosmetics manufacturing, food processing, or used as a biomaterial (with specific target tissues included for tissue engineering applications). Creating a new column in the table to present this application information would be highly advantageous.

Response 2: We are very grateful for your suggestion. The application information has been supplemented in a new column of the table and highlighted by red. We reorganized the references to correspond to the application.

Reviewer 2 Report

This manuscript by Huang et al. attempts to review different dissolution methods for silk proteins and their effects on structure and performance of the products.  This is an important topic, as silk fibres shown some remarkable characteristics, such as biocompatibility and the fact that natural silk fibres can be produced from aqueous solutions under ambient condition.  In my opinion, the manuscipt is well written, so far as it goes, but falls short for a number of reasons - including that it contains little or no new information, compared with a recent publication by Wang et al.

1) On L38-39, the authors claim that 'Silk fibroin devoid of sericin remains insoluble in water due to its high concentration of hydrophobic amino acids, such as alanine, on its surface.'  A similar allusion is made at L61.  I do not believe that is true.   Work by Privalov and Makhatadze (e.g. J. Mol. Biol. 1993, 232, 660-679) suggested that all amino acids are hydrophilic due to their polar peptide groups.  

Describing fibroin as hydrophobic is a misunderstanding of the work by Kyte and Doolittle (J. Mol. Biol. 1982, 157, 105-132), which derived a 'hydropathy index' based largely on the probability of finding specific amino acids in the core or at the surface of globular proteins.  In that scheme, Gly, Ala and Ser (the most common amino acids in silk fibroin) were given relatively small values, suggesting no great affinity or opposition to solvation by water.  That assessment, however, may be inappropriate for an essentially random coil protein, such as fibroin in native silk feedstock.

Consequently, as a minimum, the authors should quote suitable references supporting their claim.  In that respect, refs. 17 and 20 are not the most suitable; the paper by Jin and Kaplan (Nature, 2003, 424, 1057-1061) would be more suitable.

Considering the issue of solubility, a relevant view can be drawn from the solubilities in water of sugars and polysaccharides.  Glucose is highly water soluble and several glucose-based polysaccharides (e.g. starch and pullulan) are also water soluble, yet cellulose is not.  I suggest that is due to the energetically stable crystals that are formed by cellulose, and a similar situation may also occur with silks.

2) In spite of the title and the claim made in the Abstract, the properties of silk materials obtained by way of the various solvent systems were not explored comprehensively, being limited to the brief comments given in the tables.  There is little or no attention given to changes in the protein due to degradation during degumming or dissolution and how that affects processing, mechanical behaviour or other characteristics of the products.  

3) The relevant literature has not been reviewed thoroughly.  For example, a recent work by H-Y Wang et al. Dissolution and processing of silk fibroin for materials science, Crit. Rev. Biotechnol. 2021, 41, 406-424, https://doi.org/10.1080/07388551.2020.1853030) was not cited.  

Indeed, if it were included, it would raise the question as to whether this subsequent review is actually required, as it conveys no new information.

The quality of english was OK.  A small point is that fibroin structure in solution is usually described as containing ß-turns (rather than ß-corners, etc).

Author Response

Dear Reviewer,

We are deeply grateful for professional comments concerning our manuscript entitled “Advances in Preparation and Properties of Regenerated Silk Fibroin” (Manuscript ID: ijms-2529320). These comments are all valuable and helpful for improving our manuscript. 

As suggested, there are several problems that need to be addressed. According to your constructive suggestions, we have made extensive corrections and supplements to our previous manuscript. The relevant revisions in the text are shown by yellow highlight. We have highlighted the revisions with three colors to salute the three reviewers. And the point-to-point response to the comments is appended to the end of this letter.

We are looking forward to hearing from you regarding our revised manuscript. We would be glad to respond to any further questions and comments that you may have.

Yours sincerely,

Qing Zhang

Response of reviews comment

Comments:

This manuscript by Huang et al. attempts to review different dissolution methods for silk proteins and their effects on structure and performance of the products.  This is an important topic, as silk fibres shown some remarkable characteristics, such as biocompatibility and the fact that natural silk fibres can be produced from aqueous solutions under ambient condition.  In my opinion, the manuscipt is well written, so far as it goes, but falls short for a number of reasons - including that it contains little or no new information, compared with a recent publication by Wang et al.

Point 1: On L38-39, the authors claim that 'Silk fibroin devoid of sericin remains insoluble in water due to its high concentration of hydrophobic amino acids, such as alanine, on its surface.'  A similar allusion is made at L61.  I do not believe that is true.   Work by Privalov and Makhatadze (e.g. J. Mol. Biol. 1993, 232, 660-679) suggested that all amino acids are hydrophilic due to their polar peptide groups.  

Describing fibroin as hydrophobic is a misunderstanding of the work by Kyte and Doolittle (J. Mol. Biol. 1982, 157, 105-132), which derived a 'hydropathy index' based largely on the probability of finding specific amino acids in the core or at the surface of globular proteins.  In that scheme, Gly, Ala and Ser (the most common amino acids in silk fibroin) were given relatively small values, suggesting no great affinity or opposition to solvation by water.  That assessment, however, may be inappropriate for an essentially random coil protein, such as fibroin in native silk feedstock.

Consequently, as a minimum, the authors should quote suitable references supporting their claim.  In that respect, refs. 17 and 20 are not the most suitable; the paper by Jin and Kaplan (Nature, 2003, 424, 1057-1061) would be more suitable.

Considering the issue of solubility, a relevant view can be drawn from the solubilities in water of sugars and polysaccharides.  Glucose is highly water soluble and several glucose-based polysaccharides (e.g. starch and pullulan) are also water soluble, yet cellulose is not.  I suggest that is due to the energetically stable crystals that are formed by cellulose, and a similar situation may also occur with silks.

Response 1: We deeply apologize for our inaccurate representation. We have carefully read the literatures mentioned in the comment and re-written this part. In the revised manuscript, we have explained the low solubility of silk fibroin(SF) in aqueous solution at the structural level. The highly repetitive Gly-X (X is Ala, Ser, Thr, Val) dipeptide motifs of the SF heavy chain order themselves to form a microcrystalline structure through the intramolecular/intermolecular forces, such as hydrogen bonding, van der Waals forces, hydrophobic interactions.1-2 This crystalline structure allows SF to stabilize in aqueous solution. We are appreciated for your sincere suggestions.

Point 2: In spite of the title and the claim made in the Abstract, the properties of silk materials obtained by way of the various solvent systems were not explored comprehensively, being limited to the brief comments given in the tables.  There is little or no attention given to changes in the protein due to degradation during degumming or dissolution and how that affects processing, mechanical behaviour or other characteristics of the products.  

Response 2: We are extremely grateful for you to point out these problems. We have complemented the compositional and conformational changes of silk fibroin caused by various extraction processes. The characters of RSF obtained through various solvent systems are refined in the related text section. Section 3 about SF degumming was also added to try to explain the changes of SF during degumming. And we also try to discuss the RSF products properties including processability, mechanical properties, biocompatibility, et al.. We also try to pay some attentions to the influence of RSF features on RSF processability and properties of RSF products. These supplements are marked with yellow highlight.

Point 3: The relevant literature has not been reviewed thoroughly.  For example, a recent work by H-Y Wang et al. Dissolution and processing of silk fibroin for materials science, Crit. Rev. Biotechnol. 2021, 41, 406-424, https://doi.org/10.1080/07388551.2020.1853030) was not cited.  

Indeed, if it were included, it would raise the question as to whether this subsequent review is actually required, as it conveys no new information.

Response 3: We are very sorry for our negligence in reviewing the literature. We have carefully read the review and some related reviews on silk protein processing3-4. These excellent reviews listed the preparation process of silk fibroin in details, the RSF forming process and the properties of RSF products. None of them has summarized the molecular weight, conformation, processability, and stability of RSF and the influence of characters of the prepared RSF on forming process and properties of RSF products. Our manuscript could serves as a complement to the above reviews by establishing a systematic link between the dissolution process and the structure and characters of RSF, and providing valuable references for future forming process, research and the application of RSF in diverse fields.

Point 4: The quality of english was OK. A small point is that fibroin structure in solution is usually described as containing β-turns (rather than β-corners, etc).

Response 4: We are very sorry for the mistakes in this manuscript. We have revised it into “ β-turns”. We have checked the whole manuscript.

  1. Cong-Zhao Zhou, Fabrice Confalonieri, Nadine MedinaYvan Zivanovic, Catherine Esnault, Tie YangMichel Jacquet, Joel Janin, Michel DuguetRoland Perasso,  Zhen-Gang Li, Nucleic Acids Res, 2000, 28(12),2413-2419.
  2. Cong-Zhao ZhouFabrice ConfalonieriMichel JacquetRoland PerassoZhen-Gang LiJoel Janin,Proteins, 2001, 44(2), 119-122.
  3. Zongpu Xu, Weiwei Gao, Hao Bai ,iScience. 2022, 25(3): 103940.
  4. Jugal Kishore Sahoo, Onur Hasturk, ThomasFalcucci, David L.Kaplan, Nature reviews, Chemistry, 2023, 7(5).

Reviewer 3 Report

The review manuscript entitled “Advances in Preparation and Properties of Regenerated Silk Fibroin” explores and summarises different dissolution methods of silk fibroin. As mentioned by the authors, this review will be valuable for researches interested in utilizing regenerated silk fibroin in diverse fields. I have some comments as listed below:

11. There are two places where authors mentioned “random curl” (lines 77 and 128). Is it supposed to be “random coil”? Please check.

22. I understand that this review focus is about SF dissolution method. However, it is known that degumming process will also affect Mw of final RSF. It might help readers if this is mentioned including common degumming conditions (temp, time, etc).

33. Table 3 does not contain Mw information. If these are available, please add.

44. About references in Table 5, is there any dialysis step? It is mentioned that resulting RSF is resistant to gelation. Is it because ionic liquid is present? If dialysis step is unnecessary for ionic liquid, it should be mentioned as an advantage.    

Author Response

Dear Reviewer,

Thank you very much for your constructive comments concerning our manuscript entitled “Advances in Preparation and Properties of Regenerated Silk Fibroin” (Manuscript ID: ijms-2529320). These comments are very helpful for us to revise the manuscript.

Based on the suggestions and problems mentioned in your review report, we have carefully revised our manuscript. The relevant revisions are marked in green highlight in the text. We have highlighted the revisions with three colors to salute the three reviewers. And the point-to-point response to the comments is attached at the end of this letter.

We deeply appreciate your positive comments and valuable suggestions. Looking forward to your reply about my revised paper.

Yours sincerely,

Qing Zhang

Response of reviews comment

Comments:

The review manuscript entitled “Advances in Preparation and Properties of Regenerated Silk Fibroin” explores and summarises different dissolution methods of silk fibroin. As mentioned by the authors, this review will be valuable for researches interested in utilizing regenerated silk fibroin in diverse fields. I have some comments as listed below:

Point 1: There are two places where authors mentioned “random curl” (lines 77 and 128). Is it supposed to be “random coil”? Please check.

Response 1: We apologize for the language problems in the manuscript. We have revised it into “random coil”. We have checked the whole manuscript.

Point 2: I understand that this review focus is about SF dissolution method. However, it is known that degumming process will also affect Mw of final RSF. It might help readers if this is mentioned including common degumming conditions (temp, time, etc).

Response 2: Thank you for your suggestion very much. We have added Section 3 SF degumming to clarify the effect of the degumming process on RSF (from line 106 to 128).

Point 3: Table 3 does not contain Mw information. If these are available, please add.

Response 3: We are very appreciated for your suggestion and kindness, and we are very sorry for our careless. We again retrieved and read carefully the literatures about salt-acid systems, and did not find any report about the molecular weight of RSF retracted by salt- acid. Researchers possibly ignore the molecular weight of RSF. We have added the molecular weights to Table 3 and explained the situation in lines 233 - 234.

Point 4: About references in Table 5, is there any dialysis step? It is mentioned that resulting RSF is resistant to gelation. Is it because ionic liquid is present? If dialysis step is unnecessary for ionic liquid, it should be mentioned as an advantage.

Response 4: We are grateful for the mentioned problems and the suggestions. When processing with ionic liquids, there is no dialysis step and the RSF is separated from the ionic liquid by the addition of the polar organic solvent such as methanol, ethanol, isopropanol and so on. The polar organic solvent can be further removed for RSF precipitate by volatilization. At the same time the ionic liquid can be recycled.1  We have added this as an advantage to the text (lines 354-363). The resulting RSF solution is not pure RSF solution, and it is contains ionic liquids. The resulting RSF solution is resistant to gelation due to formation of hydrogen bond network between ionic liquids and SF molecules. We have further clear that in the manuscript. 2  Thank you for your suggestion

  1. Hang HengQianqian DengYipeng YangFang Wang,International Journal of Molecular Sciences, 2022, 23(15).
  2. Guzmán CarissimiCesare M BaronioMercedes G MontalbánGloria VílloraAndreas Barth,Polymers (Basel), 2020, 12(6).

Round 2

Reviewer 2 Report

I would like to thank the authors for the additional work they have put into this manuscript, in the form of new passages added to the text.  While this represents a considerable improvement over the original submission, I am not convinced that the revised manuscript is ready for publication, yet.  My reservations are as follows:

1) As the authors state in their Abstract: 'silk fibroin (SF) has gained significant attention in various fields, such as biomedicine, tissue engineering, food processing, photochemistry and biosensing, owing to its remarkable biocompatibility, machinability, and chemical modifiability'.  That has resulted in a huge number of publications in recent years.  For example, Web of Science reported over 2700 results since 2013, based on the search terms 'silk' and 'solvent'.  Clearly, reviewing all of these publications is a major task.

The authors claim that their work 'comprehensively explores and summarizes different dissolution methods of SF while examining their effects on the structure and performance of RSF'.  In my opinion, however, the present text fall considerably short of that aspiration.

2) Linked to my first point, I found the paucity of references rather disappointing.  For example:

Lines 104-105, the authors state that 'Na2CO3 is the most common degumming reagent', but support that claim with only a single reference.  

Similarly, the authors allude to the mechanism and advantages of urea-based degumming on lines 108-110, but support that with only a single reference (while web of science gave 10 articles using this method, since 2013).

L149: The authors state 'Formic acid is commonly mixed with phosphoric acid [57] or inorganic salts [58].'  Again, if these are common approaches, I suspect there should be more than 1 reference for each.

Moreover, in this case, it is also not clear what 'inorganic salts' have been investigated.

3) In the worst cases, several claims are made without any references.  Some examples include:

L32-35: 'Current evidence suggests that the molecular weight and protein structure of RSF vary depending on the extraction method used, consequently impacting its mechanical strength, degradability, and thermal stability, which in turn affect the processing and application of RSF materials.'

L179-182: 'High concentrations of neutral salts, such as LiBr, LiSCN, CaCl2, and Ca(NO3)2, under hydrothermal conditions, can dissolve SF. The presence of lithium salts in solution disrupts the intermolecular hydrogen bonds and van der Waals forces within SF due to the strong polarity of the ions, promoting the dissolution of SF.'

L200-202: 'In a calcium salt solution, SF fibers swell, and Ca2+ ions penetrate SF, complexing with tyrosine and serine side chain groups. This causes a change in the SF molecular structure from ß-folded to a random coil/a-helical conformation.'

264-266:  '...the CaCl2-ethanol-water ternary system has become one of the most commonly used methods for SF dissolution due to its low cost, fast dissolution rate, high productivity, harmlessness to organisms, and environmental friendliness.'

L297-300: 'Due to the weak solubilization of LiBr and the unique structure of amphiphilic blocks in SF molecules, which have an inherent tendency for ß-folding, SF molecular chains tend to extend and aggregate at high shear rates, resulting in shear thickening behavior.'

L385-387: 'When SF is dissolved by acid or alkali reagents, the tryptophan, serine, and threonine residues of SF are partially destroyed, reducing the glossiness of RSF fibers.'

The authors should support their claims with suitable references.

4) Some statements should be explained more clearly.  For example, at lines 342-344, the authors state that
'Ionic liquids can disperse the filament components well in aqueous solution through the formation of hydrogen bonds between them, and thus the resulting RSF solution containing ionic liquids is stable, resistant to gelation, and suitable for storage'. In fact, the work by Carissimi et al. (ref. 116) suggests that some water can be added to a SF solution in 1-ethyl-3-methylimidazolium acetate (EmimAc) without obvious precipitation, or ß-sheet structure being observable by infrared spectroscopy - but this reference does not explore how much water can be added.

5) Some statements were confusing and, potentially, wrong.  For example:

193-195:  The authors state that 'LiSCN, on the other hand, can dissolve SF at room temperature without causing degradation of SF heavy chains.'  In fact, data in ref. 69 did not suggest that silk can be dissolved in LiSCN solution without degradation.  The SDS-PAGE data (Fig. 2 in ref. 69) showed very polydispersed molecular weights, suggesting considerable degradation.

It is also of importance that ref. 69 used Antheraea pernyi, rather than Bombyx mori silk, which might affect the dissolution conditions required, but would probably not affect any chemical changes occurring.

L261-262: The authors state that 'alcohol molecules can enter the hydrophobic regions of the filamentous protein chains, rendering them hydrophilic'  Are the authors suggesting that swelling the protein can change its chemical affinity for different solvents?  Please explain.

There were also a number of relatively trivial issues that, nevertheless, make it harder to read the text:

6) The layout of the tables should be improved (e.g. consider using landscape orientation and adding horizontal lines to separate the text), so that the comments relating to different solvent systems can be distinguished more clearly.

L295 and 300: Please explain what is meant by 'solventization'.  (Do the authors mean solvation or solvent strength?)

For these reasons, I do not believe the present manuscript meets the standards required for publication.

Generally, the standard of English was good, although I noticed a few typographical errors requiring correction.

Author Response

Dear Reviewer,

Thank you very much for the constructive comments concerning our revised manuscript entitled “Advances in Preparation and Properties of Regenerated Silk Fibroin” (Manuscript ID: ijms-2529320). This review report is very detailed to point out the problems in our manuscripts, and we really appreciate the time and effort you put into this.

Based on your suggestions, we have carefully corrected those inaccurate statements in the manuscripts and the modified statements are shown using green highlight. In response to inadequate literature, we have carefully examined the manuscript and supplemented more appropriate references. These additional references are highlighted by yellow. Besides, a few new contents have been added to the text, which are marked with blue highlight. The detailed point-to-point response to the comments is listed below this letter.

Thanks again for your reviews of this manuscript and your constructive comments. We are deeply looking forward to your favorable decision.

Yours sincerely,

Qing Zhang

Response of review’s comment

Comments:

I would like to thank the authors for the additional work they have put into this manuscript, in the form of new passages added to the text.  While this represents a considerable improvement over the original submission, I am not convinced that the revised manuscript is ready for publication, yet.  My reservations are as follows:

Point 1: As the authors state in their Abstract: 'silk fibroin (SF) has gained significant attention in various fields, such as biomedicine, tissue engineering, food processing, photochemistry and biosensing, owing to its remarkable biocompatibility, machinability, and chemical modifiability'.  That has resulted in a huge number of publications in recent years.  For example, Web of Science reported over 2700 results since 2013, based on the search terms 'silk' and 'solvent'.  Clearly, reviewing all of these publications is a major task.

The authors claim that their work 'comprehensively explores and summarizes different dissolution methods of SF while examining their effects on the structure and performance of RSF'.  In my opinion, however, the present text fall considerably short of that aspiration.

Response 1: We are deeply sorry for our imperfect work. We have reviewed those associated publications and added many classical and representative references on different dissolution systems in our manuscript. They were highlighted by yellow.

Point 2: Linked to my first point, I found the paucity of references rather disappointing.  For example:

Lines 104-105, the authors state that 'Na2CO3 is the most common degumming reagent', but support that claim with only a single reference.  

Similarly, the authors allude to the mechanism and advantages of urea-based degumming on lines 108-110, but support that with only a single reference (while web of science gave 10 articles using this method, since 2013).

L149: The authors state 'Formic acid is commonly mixed with phosphoric acid [57] or inorganic salts [58].'  Again, if these are common approaches, I suspect there should be more than 1 reference for each.

Moreover, in this case, it is also not clear what 'inorganic salts' have been investigated.

Response 2: We are extremely grateful for you to point out these problems. As suggested by you, we have added more references to support these states in revised manuscript. We also carefully checked the manuscript and added references to statements with the same problems.

We have already listed the inorganic salt/formic acid systems investigated in the manuscript, and the relevant states are added in lines 167-169 & 292-295 of the text.

Point 3:  In the worst cases, several claims are made without any references.  Some examples include:

L32-35: 'Current evidence suggests that the molecular weight and protein structure of RSF vary depending on the extraction method used, consequently impacting its mechanical strength, degradability, and thermal stability, which in turn affect the processing and application of RSF materials.'

L179-182: 'High concentrations of neutral salts, such as LiBr, LiSCN, CaCl2, and Ca(NO3)2, under hydrothermal conditions, can dissolve SF. The presence of lithium salts in solution disrupts the intermolecular hydrogen bonds and van der Waals forces within SF due to the strong polarity of the ions, promoting the dissolution of SF.'

L200-202: 'In a calcium salt solution, SF fibers swell, and Ca2+ ions penetrate SF, complexing with tyrosine and serine side chain groups. This causes a change in the SF molecular structure from ß-folded to a random coil/a-helical conformation.'

264-266:  '...the CaCl2-ethanol-water ternary system has become one of the most commonly used methods for SF dissolution due to its low cost, fast dissolution rate, high productivity, harmlessness to organisms, and environmental friendliness.'

L297-300: 'Due to the weak solubilization of LiBr and the unique structure of amphiphilic blocks in SF molecules, which have an inherent tendency for ß-folding, SF molecular chains tend to extend and aggregate at high shear rates, resulting in shear thickening behavior.'

L385-387: 'When SF is dissolved by acid or alkali reagents, the tryptophan, serine, and threonine residues of SF are partially destroyed, reducing the glossiness of RSF fibers.'

The authors should support their claims with suitable references.

Response 3: We are very sorry for our negligence in writing the manuscript. We have added the appropriate references for these statements. We also scrutinized the whole manuscript and supplemented the references for the same lacking reference statements. These added references are highlighted by yellow.

Point 4: Some statements should be explained more clearly.  For example, at lines 342-344, the authors state that 'Ionic liquids can disperse the filament components well in aqueous solution through the formation of hydrogen bonds between them, and thus the resulting RSF solution containing ionic liquids is stable, resistant to gelation, and suitable for storage'. In fact, the work by Carissimi et al. (ref. 116) suggests that some water can be added to a SF solution in 1-ethyl-3-methylimidazolium acetate (EmimAc) without obvious precipitation, or ß-sheet structure being observable by infrared spectroscopy - but this reference does not explore how much water can be added.

Response 4: We sincerely appreciate the valuable comments. We have further explained the mentioned statements as follows and highlighted the added reference.

“Due to the high solvation capacity of ionic liquids, mixtures of SF and ionic liquids have high stability and low gelation rates 1. Adding a certain percentage of water to the mixture can reduce the viscosity of the solution without precipitation 2. However, the solvation of ionic liquids decreases as the water content increases 1. When the water exceeds a certain proportion, the ionic liquid/water becomes a poor solvent for RSF and RSF rebuilds β-sheet structures through hydrophobic interactions and inter- or intramolecular hydrogen bonding, thus facilitating the sol-to-gel transition 1.”

Point 5: Some statements were confusing and, potentially, wrong.  For example:

1) 193-195:  The authors state that 'LiSCN, on the other hand, can dissolve SF at room temperature without causing degradation of SF heavy chains.'  In fact, data in ref. 69 did not suggest that silk can be dissolved in LiSCN solution without degradation.  The SDS-PAGE data (Fig. 2 in ref. 69) showed very polydispersed molecular weights, suggesting considerable degradation.

It is also of importance that ref. 69 used Antheraea pernyi, rather than Bombyx mori silk, which might affect the dissolution conditions required, but would probably not affect any chemical changes occurring.

2) L261-262: The authors state that 'alcohol molecules can enter the hydrophobic regions of the filamentous protein chains, rendering them hydrophilic'  Are the authors suggesting that swelling the protein can change its chemical affinity for different solvents?  Please explain.

Response 5:

1) We are very sorry for the mistakes made in citing the literatures. In fact, we would like to quote “ Hiromi, Y et al. ,Preparation of undegraded native molecular fibroin solution from silkworm cocoons, Materials Science & Engineering C, 2001, 14(1-2), 41-46, https://doi.org/10.1016/S0928-4931(01)00207-7”.

In this study 3, LiSCN was used to solubilize raw silk fibers of Bombyx mori without alkaline degumming. The SDS-PAGE data of the resulting RSF solution showed very distinct molecular weight bands with 25 kDa and 350 kDa, similar to those of natural silk fibroin proteins. This indicates that LiSCN caused little degradation of SF.

We also re-stated the study of ref.69 (Wei Tao, et al., Structure and properties of regenerated Antheraea pernyi silk fibroin in aqueous solution, Int J Biol Macromol, 2007, 40(5), 472-478, DOI: 10.1016/j.ijbiomac.2006.11.006) of original manuscript in our revised manuscript. The RSF solutions of Antheraea pernyi obtained by NaCO3 degumming and LiSCN solubilization showed very dispersed molecular bands ranging from 20 kDa to 150 kDa, which may be due to NaCO3 degumming resulting in larger degradation of SF 4.

In addition, we have added some new contents related to the LiSCN dissolution system in the lines 222-231 of the text. 

2) We apologize for any misunderstanding caused by our inaccurate statements. Swelling does not change the chemical affinity of SF for different solvents, and it simply facilitate the penetration of salt ions. Alcohol molecules act as permeation accelerators in the dissolution process 5. We have made corrections in the text in the lines 307-309.

Point 6: There were also a number of relatively trivial issues that, nevertheless, make it harder to read the text:

1) The layout of the tables should be improved (e.g. consider using landscape orientation and adding horizontal lines to separate the text), so that the comments relating to different solvent systems can be distinguished more clearly.

2)  L295 and 300: Please explain what is meant by 'solventization'.  (Do the authors mean solvation or solvent strength?)

For these reasons, I do not believe the present manuscript meets the standards required for publication.

Response 6:

1) Thank you for your suggestion very much. We have changed the tables to the landscape orientation and enlarged the spacing between each row, making the tables more clear.

2) We apologize for the incorrect use of nouns. We want to express the content of solvation. In this manuscript, solvation refers to the interaction between solvent molecules and SF molecules, and the intensity of solvation affects the conformational stability of SF in solution.6 We have changed all “solventization” into “solvation” in the manuscript.

1. Cheng Zhang, Xin Chen, Zhengzhong Shao, ACS Biomater Sci Eng, 2016, 2(1), 12-18.

2. Maneesh K Gupta1, Shama K Khokhar, David M Phillips, Laura A Sowards, Lawrence F Drummy, Madhavi P Kadakia, Rajesh R Naik, 2007, 23(3), 1315-1319.

3. Yamada, H, Nakao, H, Takasu, Y, Tsubouchi, K, Mat Sci Eng C-mater, 2001, 14(1-2), 41-46.

4. Wei Tao, Mingzhong Li, Chunxia Zhao, Tao, Int J Biol Macromol, 2007, 40(5), 472-478.

5. Ajisawa, A., Japanese Journal of Silkworm Science, 1998, 67(2), 91-94.

6. Lin Ma, Weiren He, Aimin Huang, Lishuo Li, Zhangfa Tong, Qiaona Wei, Zilun Huang, Spectroscopy and Spectral Analysis, 2010, 30(11), 3047-3051.

Round 3

Reviewer 2 Report

I thank the authors for revising their manuscript in line with my previous comments.  In my opinion, the latest manuscript makes a good contribution to the field and merits publication.